# Gut Microbiota and Dietary Intake of Normal-Weight and Overweight Filipino Children

**DOI:** 10.3390/microorganisms8071015

**Published:** 2020-07-08

**Authors:** Maria Julia Golloso-Gubat, Quinten R. Ducarmon, Robby Carlo A. Tan, Romy D. Zwittink, Ed J. Kuijper, Jacus S. Nacis, Noelle Lyn C. Santos

**Affiliations:** 1Department of Science and Technology-Food and Nutrition Research Institute, Taguig City 1631, Philippines; robbycarlotan@gmail.com (R.C.A.T.); jaxnacis@gmail.com (J.S.N.); noellelynsantos@gmail.com (N.L.C.S.); 2Center for Microbiome Analyses and Therapeutics, Leiden University Medical Center, Albinusdreef 2, 2333 ZA Leiden, The Netherlands; q.r.ducarmon@lumc.nl (Q.R.D.); r.d.zwittink@lumc.nl (R.D.Z.); e.j.kuijper@lumc.nl (E.J.K.)

**Keywords:** children, diet urbanization, gut microbiota, dietary fiber, Filipino, nutrition

## Abstract

Diet and body mass index (BMI) have been shown to affect the gut microbiota of children, but studies are largely performed in developed countries. Here, we conducted a cross-sectional investigation on the differences in the bacterial gut microbiota between normal-weight and overweight urban Filipino children, and determined the relationship between their energy, macronutrient and dietary fiber intakes, and their gut microbiota composition and diversity. Forty-three children (normal-weight, *n* = 32; overweight, *n* = 11) participated in the study. Energy and fiber intakes were collected using a semi-quantitative Food Frequency Questionnaire (FFQ). The gut microbiota was profiled using 16S rRNA gene amplicon sequencing of the V3–V4 region. The diet of the children was a mixture of traditional and Western patterns. There were no significant differences in energy, macronutrients and energy-adjusted fiber intakes between the normal-weight and overweight groups, but there were significantly more children meeting the recommended fiber intake in the overweight group. Alpha and beta bacterial diversities did not significantly differ between weight groups. Relative abundance of *Bifidobacterium*, *Turicibacter* and *Clostridiaceae 1* were higher in the normal-weight than overweight children, and *Lachnospira* was higher in overweight children.

## 1. Introduction

The human gastrointestinal tract consists of a diverse ecosystem of microorganisms collectively known as the gut microbiota. Its composition is influenced by several factors like age, ethnicity, drugs and diet [1,2,3].

Shifts from traditional diets (high-carbohydrate, high-fiber, low-protein, low-fat) to Western diets (low-carbohydrate, low-fiber, high-protein, high-fat) have been shown to influence gut microbiota composition and diversity [4,5]. Western diets are characterized by increased intake of fast-food and food additives (e.g., emulsifiers and artificial sweeteners). It is postulated that consumption of these substances not only depletes the microbial ecosystem, but also alters the ratio of gut bacterial taxa [6].

Multiple cohort studies indicate that food patterns impact the gut microbiota in children. Children in a rural area of Burkina Faso whose diets predominantly consisted of complex carbohydrates harbored a microbiota depleted of *Firmicutes* but enriched in *Bacteroidetes* and taxa known to degrade cellulose and xylan. In contrast, the microbiota of Italian children, with diets high in animal protein and fat and low in fiber, consisted primarily of *Firmicutes* and *Proteobacteria* [4]. In a cohort of Asian children, the high abundance of *Bifidobacterium* was attributed to the carbohydrate-based diet, which is common in Asia [7].

Obesity in children has been demonstrated to affect the gut microbiota. A prospective study investigating gut microbial shifts with changes in body mass index (BMI) (i.e., from normal to a state of obesity) reported that the onset of obesity was associated with a reduction in gut microbiota diversity and a shift in composition towards an increased relative abundance of *Bacteroideceae* and a lower relative abundance of *Prevotellaceae* [8]. Taken together, these findings demonstrate the importance of diet and nutritional status in determining gut microbiota composition.

Studies on children’s microbiota were performed mostly in ‘Western’ populations in recent years, in countries with similar profiles of lifestyle and industrialization such as Israel, Japan, and China [9,10,11,12]. Apparently, the extent to which diet and BMI influence the gut microbiota of Asian children is less investigated. Initial studies characterized the diet and gut microbiota profile of children in selected Asian countries [5,7] based on enterotype classification. Accordingly, the gut microbiota of Asian children clusters into two enterotypes driven by *Prevotella* (P-type) or *Bifidobacterium*/*Bacteroides* (BB-type), and the tradeoff between these dominant taxa is diet-dependent [7].

In the Philippines, a single published study examined the impact of diet on children’s gut microbiota and indicated that urban children consuming Western diets harbored BB-type gut microbiota, whereas rural children consuming traditional diets mainly harbored P-type gut microbiota [5]. It was also noted that the relative abundance of *Prevotella* was significantly lower in the overweight-obese children living in the urban area, as compared with the normal-underweight group who were living in rural areas [5].

These observations may indicate diet- and nutrition-associated variations in the gut microbiota of children in developing countries. The primary aim of the current study was to investigate the differences in energy and macronutrient intakes and bacterial gut microbiota between normal-weight and overweight children in Taguig City, an urbanized city in Metro Manila, Philippines. Additionally, the relationship between dietary fiber intake and the gut microbiota was investigated.

## 2. Materials and Methods

### 2.1. Ethical Considerations

This study was conducted according to the guidelines specified in the Declaration of Helsinki. The study protocol was reviewed and approved (25 April 2017) by the Food and Nutrition Research Institute Ethics Review Committee (Protocol approval number FIERC-2017-003). The research team obtained signed assent and informed consent forms from all study participants and their respective parents/legal guardians before the start of the study.

### 2.2. Study Sites

The study was conducted in three government schools in Taguig City, Philippines. Data were collected between July to September 2017. Clinical assessment to evaluate the eligibility of the participants was conducted at the Department of Science and Technology-Food and Nutrition Research Institute (DOST-FNRI), Taguig City, Philippines.

### 2.3. Study Population

Participants aged seven to eleven years old were recruited. Potential participants attended a scheduled orientation session for the discussion of the study protocol. Those who submitted signed assent and parental consent forms were invited for clinical assessment. Information on age and sex were obtained by a questionnaire. A registered nutritionist-dietitian (RND) conducted anthropometric measurements, and subsequently calculated BMI to determine nutritional status using the 2007 World Health Organization (WHO) Growth Reference BMI-for-age for children [13]. Study participants were stratified into normal-weight and overweight groups based on the BMI-for-age z-score cut-off points (for children aged 5 to 19 years old: obesity ≥ +2SD; overweight ≥ +1SD; normal-weight = −2SD to +1SD). A pediatrician-gastroenterologist conducted a physical examination and clinical assessment to determine deworming treatment, and antibiotic and probiotic intakes in the past six months. Eligible participants were those who were not diagnosed with gastrointestinal disorder, no diarrhea and not taking medications at the time of the study. For girls, only those who did not experience menarche yet were considered eligible to participate. Forty-six volunteers qualified for the study but only 43 completed their participation (Figure 1).

### 2.4. Dietary Data Collection

An 81-item semiquantitative Food Frequency Questionnaire (FFQ) was used to determine habitual food intake. Food items included in the FFQ were identified from the Food Exchange Lists (FEL) for Meal Planning [14] and commonly consumed food items of Filipino children based on the Philippines 8th National Nutrition Survey [15]. The food items were grouped based on the FEL food groupings, and the ‘snack group’ was added for familiarity purposes as this is usually consumed by school children. The FFQ was filled out by the research team’s RNDs through face-to-face interviews. In the interview, each study participant was asked to assess their consumption of a particular food item in the FFQ in the past six months. A negative response (“no”) meant that the study participant has never eaten that food item in the past six months, and this response was recorded in the questionnaire. When a participant indicated a positive response (“yes”), the interviewer further asked whether a food item was consumed on a daily, weekly, monthly or yearly basis. For each option, the study participant was requested to recall the frequency of consumption, i.e., once, twice, three times per day or once, twice, three times per week. The amount of a food item normally consumed per eating period was determined using measuring cups and spoons. A photographic food atlas was also used to aid in recall of the food items. The interview took about 20–30 min per study participant.

### 2.5. Energy, Macronutrient Dietary Fiber Assessment

The energy, macronutrient and dietary fiber content of the food items were estimated using the Philippine Food Composition Tables (FCT) database [16]. All food items were analyzed as single food items; combination dishes were broken down into ingredients to derive specific single food items. The cooked or raw form of the food items in the database were carefully identified depending on the typical form in which it is consumed. Reference portion sizes used were common household measurements, converted into weight (in grams) using the FEL [14]. A food score for each food item was calculated by multiplying the amount consumed (in grams) and the deviation from the reference weight and the frequency code. Total energy, macronutrient and dietary fiber intakes were summed up for each study participant. The percentage contribution of the dietary fiber from each food group was calculated by dividing the total dietary fiber intake from the food group over the total dietary fiber intake of the study participant. Then, the average percentage of each food group was calculated. Dietary fiber was adjusted for energy intake by adapting the residual method [17]. Participants were grouped into tertiles based on their total fiber intake in grams per day (<12.43, 12.44–16.33, >16.33).

### 2.6. Fecal Sample Collection and DNA Extraction

Pellet-sized fresh stool samples were collected from each study participant in sterile feces tubes. Stool samples were transferred to the FNRI Nutritional Genomics Laboratory within two hours after collection. Upon arrival at the laboratory, stool samples were aliquoted into 15 mL conical tubes (Falcon™, Fisher Scientific, Hampton, NH, USA) containing two mL RNAlater and ~one gram feces (Ambion, Inc., Austin, TX, USA). The aliquot was stored at −20 °C until DNA extraction. Genomic DNA was extracted using the QIAamp^®^ DNA Stool Mini Kit (Qiagen, Hilden, Germany). The extraction protocol was based on the manufacturer’s instructions using the QIAamp DNA Stool Handbook (Qiagen, 2012). Briefly, 200 mg of stool sample was used as the primary material for the extraction. The sample was lysed with the combination of Buffer ASL and a high temperature (70 °C). After lysis, the DNA-damaging substances and PCR inhibitors present in the sample were adsorbed to the InhibitEX matrix. The matrix was then pelleted by centrifugation and the DNA in the supernatant was purified on QIAamp Mini Spin Columns. The DNA yield and overall quality were measured using Epoch™ Microplate Spectrophotometer (Biotek, Winooski, VT, USA).

### 2.7. Library Preparation and 16S rRNA Gene Amplicon Sequencing

The Next Generation Sequencing (NGS) library preparations and Illumina MiSeq (San Diego, CA, USA) sequencing were conducted at First BASE Laboratories (Selangor, Malaysia). The quantity and quality of the PCR products that targeted the V3 and V4 regions were measured using Tapestation 4200 (Agilent Technologies, Santa Clara, CA, USA), PicoGreen™ (Molecular Probes, Eugene, OR, USA), and NanoDrop instruments (Life Technologies, Grand Island, NY, USA). All samples passed the quality control measurements and proceeded to the library preparation method recommended by Illumina. Approximately 5–30 µg DNA was used to generate amplicons using a two-stage PCR, with PCR clean-up in between. The bacterial 16S rRNA was amplified using the universal primers 341F (5′-CCTAYGGGRBGCASCAG-3′) and 806R (5′-GGACTACNNGGGTATCTAAT-3′). The total reaction volume (50 µl) was composed of genomic DNA, 0.3 pmol of each primer, 1 unit of Toyobo KOD Multi Epi DNA (Osaka, Japan), and Toyobo 2x KOD Multi Epi PCR buffer containing dNTPs and water (Osaka, Japan). The PCR was performed as follows—1 cycle (94 °C for 2 min) for initial denaturation followed by 25 cycles (98 °C for 10 s, 51 °C for 30 s, and 68 °C for 60 s) for annealing and extension of the amplified DNA. The quality of the libraries was measured using Tapestation 4200 (Agilent Technologies, Santa Clara, CA, USA), PicoGreen™ (Molecular Probes, Eugene, OR, USA), and qPCR (Applied Biosystems, Carlsbad, CA, USA). All the samples passed the QC measurement. These libraries were pooled according to the protocol recommended by Illumina and proceed straight to sequencing using MiSeq platform at 2 × 301 paired-end (PE) format.

### 2.8. Sequencing Data Processing

Raw sequences were processed using the QIIME 2 pipeline (v2019.4.0) with the Deblur algorithm. Quality plots of raw data were inspected prior to trimming, and based on inspection of these plots, a trim length of 220 bp was chosen. Obtained OTUs were taxonomically assigned using the pretrained Naïve Bayes classifier on the Silva_132_SSU Ref database with 99% similarity. The OTU table was filtered for OTUs which made up less than 0.005% of total read count, as previously recommended [18].

### 2.9. Statistical Analysis of the Dietary Data

Data were checked for normality using visual QQ plots. Participants’ characteristics were analyzed using descriptive statistics. Continuous variables were presented as means and standard deviation or median and interquartile range while percentages are given for categorical variables. Mean difference between the normal-weight and overweight group was analyzed using the Wilcoxon rank sum test. All descriptive statistical analyses were performed using SPSS (version 22, IBM, Armonk, NY, USA).

### 2.10. Statistical Analysis of the Gut Microbiota Data

All statistical analyses were performed in R (v3.6.1) using the phyloseq (v1.28.0), microbiome (v1.6.0), DESeq2 (v1.24.0) and vegan (v2.5–5) packages. Data were collapsed at the genus level using the tax_glom function. Alpha diversity measures were calculated based on the 0.005% filtered OTU tables. To test for group differences in alpha diversity, we performed *t*-tests and one-way ANOVA. Normality and equal variances were investigated using Shapiro–Wilk test and Levene’s test. All further analyses were performed at the genus level. To test for the effect of clinical variables on the overall gut microbiota composition, permutational multivariable analysis of variance (PERMANOVA) based on Bray–Curtis dissimilarity was employed using the adonis function from the vegan package (999 permutations). Prior to employing PERMANOVA, betadisper was used to test for variance homogeneity between the group of interests. No comparisons indicated a difference in group dispersions. To visualize differences in overall gut microbiota composition (beta diversity) between clinical groups, we used PCoA based on Bray–Curtis dissimilarity. For differential abundance analysis between groups, the DESeq2 method was employed. Prior to employing DESeq2, genera not present in at least 25% of the samples were removed to minimize zero-variance errors and spurious significance. *p*-values were adjusted by employing the Benjamini–Hochberg correction.

### 2.11. Data Accessibility

All raw dietary and sequencing data used in the current study are accessible upon request.

## 3. Results

### 3.1. Study Participants

A total of 43 participants were included and were stratified in normal-weight (*n* = 32) and overweight (*n* = 11) groups. The overall characteristics of the participants are shown in Table 1. Generally, there were more female than male participants. Almost half of the study participants received deworming treatment (*n* = 20), 9 children used antibiotics and 30 had intake of probiotics in the past 6 months before the start of the study, although no statistical difference regarding these variables was observed between the weight groups.

For Filipino children 6–9 years old, the recommended energy intake is 1470 kcal/day for females, and 1600 kcal/day for males, and the adequate intake (AI) for dietary fiber should be 11–14 g/day for age 10–12 years old, the recommended energy intake is 1980 kcal/day for females, and 2060 kcal/day for males, and AI for dietary fiber should be 15–17 g/day [19]. Data from the FFQ showed that there were no significant differences in the total energy, macronutrients and energy-adjusted dietary fiber intakes between the normal-weight and overweight groups. In terms of the proportion of children meeting the recommended energy and fiber intakes between groups, only 50% of the children in the normal-weight group and 36.4% in the overweight group met the requirement for energy. Notably, the proportion of children meeting the recommended fiber intake was significantly different between groups (Table 2).

For both groups, energy intake from carbohydrates was 66% while energy from protein and fats were 12% and 22%, respectively. Contribution of macronutrients to energy intake for Filipino children are within the recommended Acceptable Macronutrient Distribution Range (AMDR) of 55–79% for carbohydrates, 6–15% for protein, and 15–30% for fat.

### 3.2. Dietary Fiber Intake by BMI

Most of the dietary fiber intake of the study participants is from consuming foods in the rice and alternatives food group. Consumption of fruits contributes more to dietary fiber intake of the participants than vegetables. The snacks food group, which consists of savory and sweet foods like hamburgers, banana cues (deep-fried banana in caramelized sugar), and sandwiches, contributes 10% to the overall dietary fiber intake of the participants; while the other food groups (i.e., legumes and nuts, sweets, meat and alternatives and dairy) have a minor contribution to overall intake of dietary fiber (Figure 2).

There is minimal difference in dietary fiber intake between the normal-weight and overweight groups. Dietary fiber intakes from fruit and snacks are slightly higher in overweight children as compared to the normal-weight children, but this difference is not statistically significant (*p* = 0.20). In terms of dietary fiber contribution of the vegetable group, an opposite observation was noted (Figure 3).

### 3.3. Sequencing Results

A total read number of 845,430 was obtained from 43 samples, with a median of 20,250 reads per sample (range 10,585–27,044) resulting in a total of 1741 OTUs after filtering on 0.005% abundance.

### 3.4. Alpha Diversity

To investigate alpha diversity, we computed both Chao1 richness and Shannon diversity indices. No statistical difference was found between the normal-weight and overweight groups for Chao1 (independent *t*-test, *p* = 0.873), nor for Shannon (*p* = 0.140). In addition, no differences were found between tertiles of energy-adjusted fiber intake for Chao1 (ANOVA, *p* = 0.674) nor Shannon (*p* = 0.158).

### 3.5. Bacterial Community Structure

At the family level, the children’s microbiota is characterized by a high relative abundance of *Bacteroidaceae, Prevotellaceae, Ruminococcaceae, Lachnospiraceae*, and *Veillonellaceae* (Figure 4). When zooming in at the genus level, it is further confirmed that the microbiota is characterized by either a high abundance of *Bacteroides* or *Prevotella 9* for most children in both normal-weight and overweight groups.

To test for difference in overall community structure between normal-weight and overweight children, we performed permutational variable analysis of variance (PERMANOVA). However, this was not significant at the OTU and genus level (*p* = 0.479 and *p* = 0.337, respectively). In addition, Bray–Curtis dissimilarities were ordinated using PCoA ordination (Figure 5). In addition, no differences were found between tertiles of energy-adjusted fiber intake at OTU (*p* = 0.254) and genus (*p* = 0.611) levels.

### 3.6. Differential Abundance Analysis

DESeq2 was used to determine whether any differences in relative abundance of taxa existed between the normal-weight and overweight group. *Bifidobacterium*, *Turicibacter* and *Clostridiaceae 1* were more abundant in the normal-weight group (*p* = 0.036, *p* = 0.036 and *p* = 0.043, respectively; Figure 6A).

Regarding fiber intake, the tertile with the thighest energy-adjusted fiber intake was compared with the tertile of lowest fiber intake. Three genera were more abundant in the highest tertile, namely *Lachnospira*, *Erysipelotrichaceae UCG-003* and an undefined member of the *Peptostreptococcaceae* family (all *p*-values < 0.01), although differences seem to be driven by very few children (Figure 6B).

## 4. Discussion

In this study, we investigated the differences in the composition and diversity of the gut microbiota of normal-weight and overweight Filipino children. We also examined the association of gut microbiota and habitual levels of dietary fiber intake. No significant differences in the gut microbial diversity and overall community structure were observed between the normal-weight and overweight children and across energy-adjusted fiber intakes. However, the difference in relative abundance of specific bacterial taxa could be observed. *Bifidobacterium*, *Turicibacter* and *Clostridiaceae 1* were more abundant in the normal-weight than overweight group. *Lachnospira*, *Erysipelotrichaceae UCG-003* and *Peptostreptococcaceae* were more abundant in the group with the highest-energy adjusted fiber intake as compared to lowest fiber intake.

It is still unclear how the proportion of *Bifidobacterium* can affect host BMI, and vice versa. Some authors argue that that imbalance of microbes (i.e., lower content of *Bifidobacterium* and higher content of *E. coli*) might impact weight status [20] and that *Bifidobacterium* can be inferred protective due to a higher relative abundance of *Bifidobacterium* in normal-weight individuals. In contrast, Sepp, et al. [21] reported a concordance between body weight and BMI and a higher proportion of both *Bifidobacterium* and *Eubacteria* among 5-year old children in Estonia. Conversely, our findings support the findings of Kalliomäki, et al. [22] and Ignacio, et al. [23] who both found a higher number of *Bifidobacterum* in children who were able to maintain a normal weight at 7 years and in lean Brazilian children aged 3–11 years, respectively.

The bacterial genus *Turicibacter* appears to be sensitive to dietary components. It increases in diets rich in protein and low in carbohydrates [24], high-resistant starch [25], or after a long-term consumption of walnuts [26]. We report the abundance of *Turicibacter* among our normal-weight participants. This finding is in contrast with the findings of Chen, et al. [27] who found an enrichment of *Turicibacter* among obese Chinese children. Meanwhile, our findings support the reduction in the relative abundance of *Clostridiaceae* after the onset of obesity among 2–9-year-old children in eight European countries [8]. Earlier studies reported that *Clostridium* declines in fecal samples of severely obese adult humans [28]. A reduced number of *Clostridiaceae* was also observed among obese participants from two independent adult populations in the United States [29].

*Lachnospira* is one of the main genera of the human gut microbiome [30,31] and is known to be beneficial for both humans and animals by fermenting fibers, thereby producing short chain fatty acids [32,33,34]. The mean fiber intake of both our normal-weight (14.3 g) and overweight (16.7 g) participants is near the recommended intake of 11–14 and 15–17 g of fiber for Filipino children aged 6–9 and 10–12 years old, respectively [19]. With a general dietary pattern composed of plant-based foods, it is noteworthy to mention that only 32.5% of our participants with normal-weight met the recommended fiber intake. In a study linking the quantitative dietary intake with microbiome features, Herman, et al. [35] observed that fiber consumption is positively associated with the relative abundance of *Lachnospira* among 2-to-9-year old children in Los Angeles, California. Similar studies conducted among adults found that a plant-based diet correlates with a relative abundance of *Lachnospira* [31] and that increased fiber intake modulates the increase in *Lachnospira* after dietary fiber supplementation [36].

To date, there is limited evidence for the association of *Erysipelotrichaceae UCG-003* and *Peptostreptococcaceae* with fiber intake. Members of the *Peptostreptococcaceae* family are likely involved in the formation of butyrate from glutamate [37] and were observed to be positively correlated with the production of branch-chained fatty acids [38]. An over-representation of *Peptostreptococcacea* was found among children residing in rural areas of Thailand who happened to consume more rice and vegetables than their urban counterparts [37].

On average, the recommended energy intake for Filipino boys and girls aged 6–9 years old is 1535 kcal/day. While the energy intake from macronutrients is within the AMDR for Filipino children, the energy intake of children in the present study is higher (13% higher in the normal-weight group and 27% higher in overweight group) than the recommended amount. Most of the energy consumed by children are from carbohydrate-rich foods like rice, bread and noodles. The dietary intake of children in the present study was in between the intake levels of rural and urban children in Leyte, Philippines. Also, we noted that the gut microbial community of children in the present study is characterized by a high relative abundance of either *Bacteroideceae* or *Prevotellaceae*, with some profiles containing a mix of these bacteria. This suggests that the gut microbial community structure of the children in our study resides in between the microbiota classification of rural and urban children in the Leyte study. In addition, it implies that the diet of children in the present study is not yet highly Westernized, but rather a mixture of traditional and Western dietary patterns. In a Nigerian study it was highlighted that while the dominance or under-representation of specific bacteria characterizes the rural versus urban diet dichotomy, the presence of a mixture of gut bacteria indicates a potential and gradual adaptation of the gut microbial community with dietary shifts (i.e., from traditional to Western diets) [39]. This, together with our observations, lends support to the view that transition in diets appear to have a gradient effect on gut microbial composition.

We acknowledge that the present study is limited due to the relatively small sample size. The study also lacks in-depth biological and biochemical analyses, thereby limiting the power to strongly link macronutrient components to genus-level gut taxonomy, while not allowing more complex assessment of the gut microbial composition relative to potential nutrient-nutrient interactions, host physiology and other factors. Furthermore, though we use visual tools to aid during the dietary collection, FFQ is prone to over- or under-reporting, which can be driven by providing a socially-acceptable response [40]. The study team used the energy-adjustment method to get the estimate of dietary fiber intake. Nonetheless, gut microbiota research in the Philippines is very limited and this study contributes important information on microbiota composition of Filipino children and the role of diet in shaping their gut microbiota. This may prove useful in future investigations on dietary interventions that could modulate the gut microbiota to optimize the health benefits conferred by the diet.

## 5. Conclusions

Our results indicate that the diet of children in the study is likely a mixture of traditional and Western patterns. There were no significant differences in the energy, macronutrients and energy-adjusted fiber intakes between the normal-weight and overweight groups, but there were significantly more children meeting the recommended fiber intake in the overweight group. For both groups, energy and fiber intakes are mostly coming from the consumption of rice and alternatives (e.g., noodles, bread), fruits, vegetables and snacks. We did not see clear differences in the overall gut microbial diversity and community structure between the normal-weight and overweight groups but we noted that the relative abundance of *Bifidobacterium*, *Turicibacter* and *Clostridiaceae 1* were higher in the normal-weight than the overweight children. *Lachnospira*, on the other hand, was higher in the overweight group, particularly among those with the highest energy-adjusted fiber intakes. This study contributes important information on the relationship between dietary components and the bacterial gut microbiota composition of Filipino children.

## Figures and Tables

**Figure 1 microorganisms-08-01015-f001:**
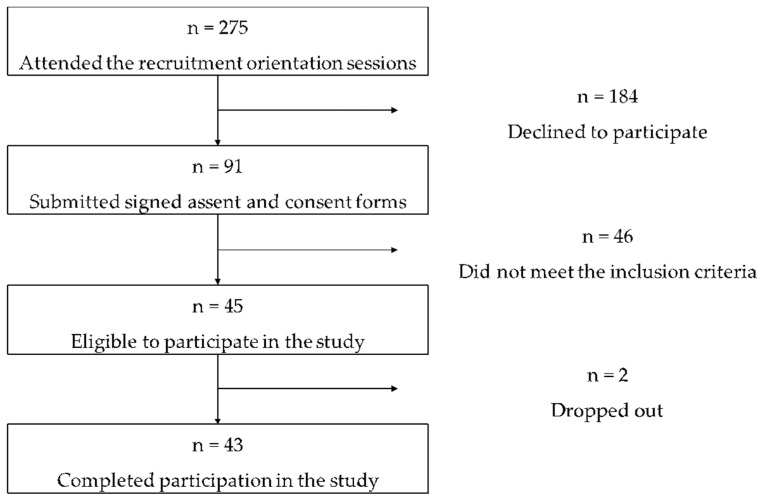
Flow diagram for recruitment, screening, selection, and retention of study participants.

**Figure 2 microorganisms-08-01015-f002:**
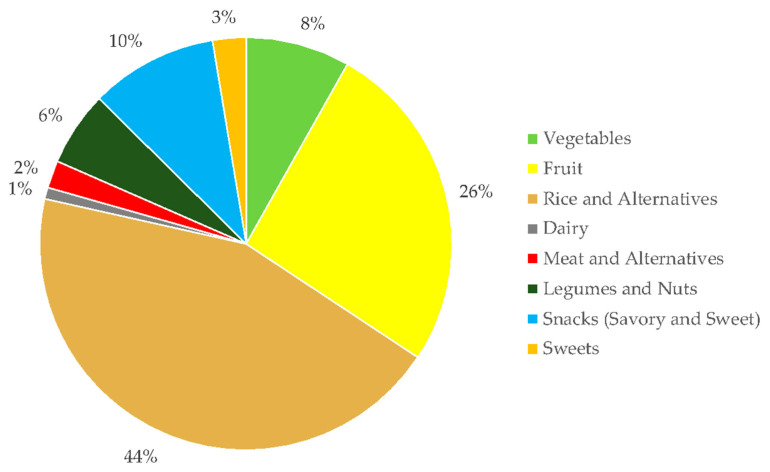
Percentage contribution of food groups to overall dietary fiber intake.

**Figure 3 microorganisms-08-01015-f003:**
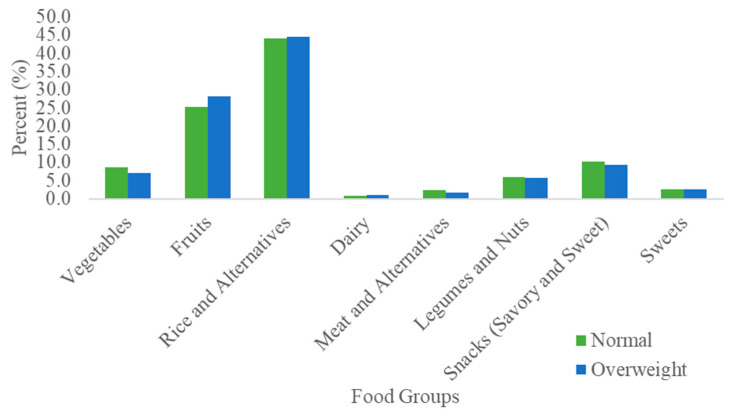
Percentage contribution of food groups to overall dietary fiber intake, by BMI group.

**Figure 4 microorganisms-08-01015-f004:**
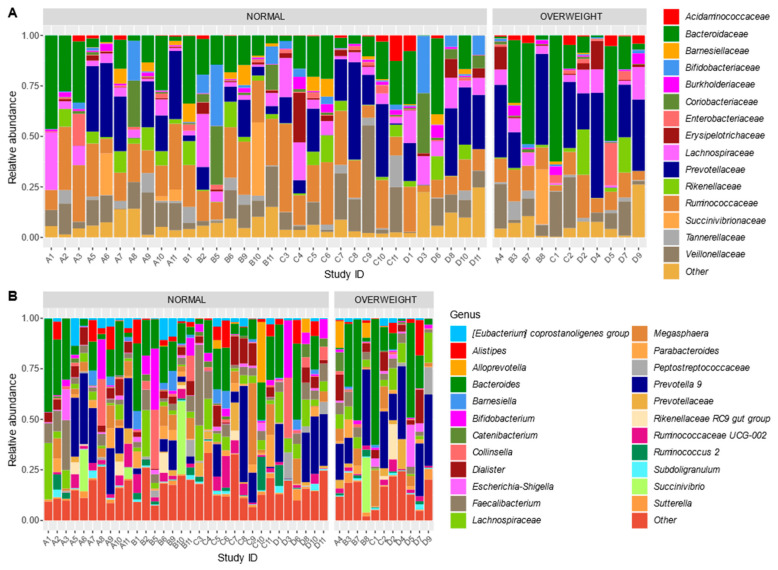
Relative abundance profiles at family (**A**) and genus (**B**) level. The category ‘Other’ indicates the sum of all other families or genera per sample.

**Figure 5 microorganisms-08-01015-f005:**
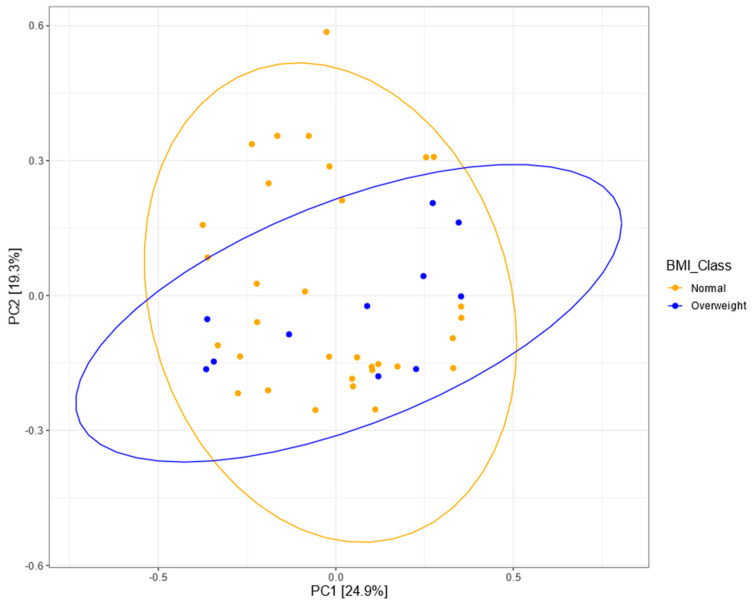
PCoA plot using Bray–Curtis dissimilarity. The 95% confidence ellipses are drawn around the BMI groups.

**Figure 6 microorganisms-08-01015-f006:**
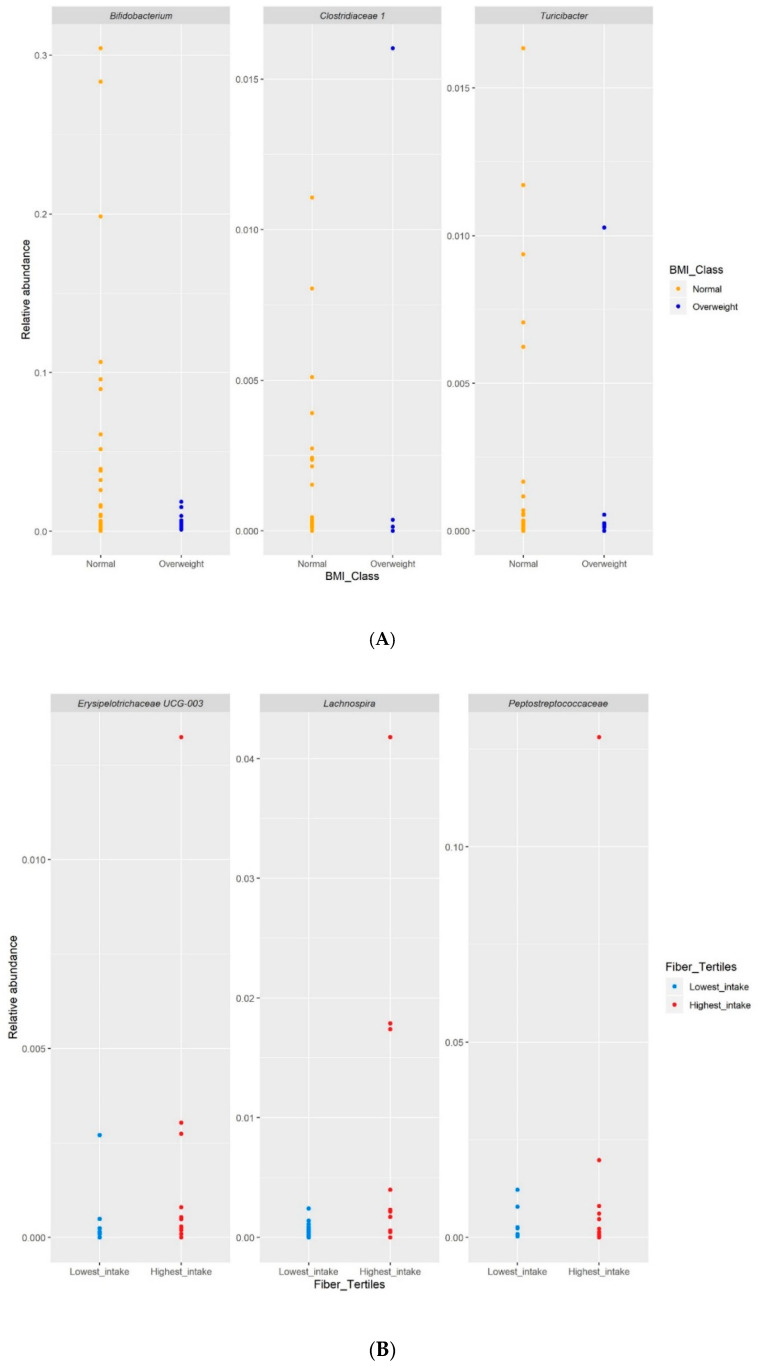
(**A**): Relative abundance of taxa between the normal-weight and overweight groups. (**B**): Differential relative abundance in genera by fiber intake tertile.

**Table 1 microorganisms-08-01015-t001:** Characteristics of study participants.

		Normal-Weight (*n* = 32)	Overweight (*n* = 11)	*p* Value *
Age in years, (SD)		9.2 (0.9)	8.6 (0.9)	0.50
Sex, *n* (%)	Male Female	11 (34) 21 (66)	5 (45) 6 (55)	
Deworming, *n* (%)	With Without	17 (53) 15 (47)	3 (27) 8 (73)	0.14
Antibiotic use in the past 6 months, *n* (%)	With Without	8 (25) 23 (72)	1 (9) 10 (91)	0.42
Probiotic intake in the past 6 months, *n* (%)	With Without	3 (9) 29 (91)	0 (0) 10 (91)	0.14

* Values shown are expressed as mean (SD) or *n* (%). *p* values were obtained by Wilcoxon rank sum test for continuous variable or X2 test for categorical variables 3.2. Energy, macronutrient and dietary fiber intakes.

**Table 2 microorganisms-08-01015-t002:** Energy, macronutrient, and dietary fiber intake profile of study participants.

	Normal-Weight (*n* = 32)	Overweight (*n* = 11)	*p*-Value *
Energy intake (kcal)	1735.0 (461)	1950 (546)	0.20
Carbohydrate intake (g)	285.5 (74.8)	322.9 (21.1)	0.20
Protein intake (g)	52.9 (16.8)	60.1 (24.7)	0.50
Fat intake (g)	42.4 (13.3)	46.6 (21.6)	0.90
Energy-adjusted dietary fiber intake (g)	14.3 (5.1)	16.7 (6.0)	0.20
Meeting the recommended energy intake, *n* (%)	16 (50.0)	4 (36.4)	0.43
Meeting the recommended fiber intake *n* (%)	12 (37.5)	8 (73.0)	0.04

* Values shown are expressed as mean (SD) or *n* (%). *p* values were obtained by Wilcoxon rank sum test for continuous variable or X^2^ test for categorical variables.

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
