# Peer review of "Gut Microbiota and Dietary Intake of Normal-Weight and Overweight Filipino Children"

_microorganisms, 2020, doi:10.3390/microorganisms8071015_

Round 1

Reviewer 1 Report

It is my pleasure to review this manuscript.

Overall, the manuscript is well written and found no clear difference in overall gut microbial diversity and community structure between normal and overweight Filipino children (7-11 years old). They also found that the relative abundance of Bifidobacterium, Turicibacter and Clostridiaceae 1 were higher in the normal-weight than the overweight children. While Lachnospira was higher in the overweight group, particularly among those with the highest energy-adjusted fiber intakes.

The following are my comments and suggestions.

  1. Add the study design and study time information in the abstract.
  2. “Bacterial diversity” needs to be specific, are they alpha and beta diversity?
  3. In line 66, “diet pattern” and in line 15, “diet fiber”, need to make it consistent in your manuscript. Which one is your main focus?
  4. In the study population flow chart, it would be helpful to provide the comparison for these not participating this study (declined or dropped out) vs. participants, therefore it might evaluate the potential selection bias.
  5. Line 193-195, how might the antibiotics and probiotic affect the final results?
  6. Results in line 214-217 might also need to be shown in a table.
  7. What analysis did the authors conduct to produce the figure 2 result? Might need a little description.
  8. For alpha diversity, why would authors pick Chao1 and Shannon Diversity, not other index, such as Simpson index? I would also suggest adding a figure for it, if possible adjusting age.
  9. For the taxa abundance analysis, did the authors apply the multiple test correction for multiple comparison? For the visualization perspective, I would recommend using LEfSe and volcano plot to show the results. See following link:

http://huttenhower.sph.harvard.edu/galaxy/

  1. When assessing the relationship between gut microbiota and overweight, and gut microbiota and diet fiber, the author might also need to adjust some covariates, such as age, gender.
  2. the discussion and conclusion sections are well written.

Author Response

Thank you for your valuable comments. Below, please see our responses on all relevant points:

Point 1: Add the study design and study time information in the abstract

Response 1: The study design (line 13) was indicated in the Abstract, as suggested. Data collection period was indicated in the manuscript (line 81-82).

***

Point 2: “Bacterial diversity” needs to be specific, are they alpha and beta diversity?

Response 2: The reviewer is correct. Indeed, we were referring to both alpha and beta diversity. This has been clarified in the manuscript (line 23).

***

Point 3: In line 66 “diet pattern” and in line 15, “diet fiber”, need to make it consistently in your manuscript. Which one is your main focus

Response 3: In line 15-16 and line 68, we revised “diet pattern” to energy, macronutrient and dietary fiber intakes to specifically characterize what particular dietary pattern is being investigated in the study. This has been consistently used in the entire manuscript.

***

Point 4: In the study population flow chart, it would be helpful to provide the comparison for those not participating in this study (declined or dropped out) vs participants, therefore it might evaluate the potential selection bias

Response 4: While all recruited participants were from the aforementioned schools and were within the age range set for the study, most of them decided to voluntarily decline. Some were not able to meet the inclusion/exclusion criteria. No data was gathered further from them in relation to this study.

***

Point 5: Line 193-195, how might antibiotics and probiotics affect the final results?

Response 5: This is very difficult to estimate, for two reasons. First, we only recorded use of antibiotics and probiotics in the preceding 6 months, but study participants were instructed not to take antibiotics and probiotics during their participation in the study, and if they had to, they needed to inform the research team about it. Second, as the number of users is quite low, it is not feasible to perform adjustments on this. Lastly, we would like to note that there were no significant differences between the two weight groups regarding use of both antibiotics and probiotics (see Table 1).

***

Point 6: Results in line 214-217 might also need to be shown in a table

Response 6: This is reflected in the last two rows in Table 2 (meeting the recommended energy intake, n (%) and meeting the recommended fiber intake, n (%))

***

Point 7: What analysis did the authors conduct to produce the Figure 2 result? Might need a little description

Response 7: Description of the analysis was added (line 127-129), and additional information was inserted to enhance clarity (line 106-108)

***

Point 8: For alpha diversity, why would authors pick Chao1 and Shannon Diversity, not other index such as Simpson Index? I would also suggest adding a figure for it, if possible adjusting age

Response 8: We believe it is more a personal preference which alpha diversity metrices are chosen, as many different ones exist. Chao1 and Shannon diversity are among the most commonly used metrics for gut microbiota analyses, but Simpson would also have been fine (as well as several other possibilities). We do not believe a figure is necessary, as there are quite some figures in the manuscript already. However, if the reviewer insists this is important, we will be happy to provide an extra supplementary figure on this.

***

Point 9: For the taxa abundance analysis, did the authors apply the multiple test correction for multiple comparison? For the visualization perspective, I would recommend using LEfSe and volcano plot to show the results. See following link:  http://huttenhower.sph.harvard.edu/galaxy

Response 9: We have indeed applied multiple test correction after DESeq2 testing, please see line 191-194 where this is detailed. We opted here for a method which models the read counts (DESeq2) instead of relative abundances (as is done in LEfSe). If the reviewer insists on getting more information about the taxa abundance analysis, we could make supplementary tables with DESeq2 output.

***

Point 10: When assessing the relationship between gut microbiota and overweight, and gut microbiota and diet fiber, the authors might also need to adjust some covariates, such as age, gender

Response 10: We would like to point out that there were no significant differences between these variables (Table 2) between the two groups. Therefore, we do not believe it is necessary to adjust for these variables.

Reviewer 2 Report

Golloso-Gubat et al. studied gut microbiota and dietary intake in two groups of Filipino children; normal-weight and overweight. The analysis method and the method of presenting the data are appropriate. Some results differed significantly between the two groups and presented important information (Fig. S1 and S2). On the other hand, there were no differences in many of the results shown in the figures (Fig. 2-5). In the abstract, important points were summarized well, but the presentation is messy and the things the authors want to argue are blurred. For example, the results described in the abstract are shown in supplementary figures. I would like to suggest the authors separate the observations more clearly as 1) significant differences between the two groups of normal-weight and overweight children and 2) the features of gut microbiota and dietary intake recognized in Filipino children. Once more, I repeat that the abstract is well organized, so the authors should reflect this in the text. The research is acceptable, so I would like the structure of the paper revised to better highlight this.

Author Response

Point 1: Some results differed significantly between the two groups and presented important information (Fig S1 and S2). On the other hand, there were no differences in many of the results shown in figures (Fig 2-5). In the abstract, important points were summarized well, but the presentation is messy and the things the authors want to argue are blurred. For example, the results described in the abstract are shown in supplementary figures.

Response 1: We agree with the reviewer that the supplementary figures present important information that support our findings to answer the objectives of the study. We have integrated these figures in the main text under the Results section (Figures 6A and 6B).

***

Point 2: I would like to suggest the authors to separate the observations more clearly as (1) significant differences between the two groups of normal-weight and overweight children, and (2) the features of gut microbiota and dietary intake recognized in Filipino children

Response 2: Indeed, methods and results are presented following this sequential manner, i.e., comparison of the (1) dietary intakes and (2) gut microbiota profile of the two groups of children (normal-weight vs overweight).